# Winter temperatures limit population growth rate of a migratory songbird

Bradley K. Woodworth[1], Nathaniel T. Wheelwright[2], Amy E. Newman[1], Michael Schaub[3] & D. Ryan Norris[1]

Understanding the factors that limit and regulate wildlife populations requires insight into demographic and environmental processes acting throughout the annual cycle. Here, we combine multi-year tracking data of individual birds with a 26-year demographic study of a migratory songbird to evaluate the relative effects of density and weather at the breeding and wintering grounds on population growth rate. Our results reveal clear support for opposing forces of winter temperature and breeding density driving population dynamics. Above-average temperatures at the wintering grounds lead to higher population growth, primarily through their strong positive effects on survival. However, population growth is regulated over the long term by strong negative effects of breeding density on both fecundity and adult male survival. Such knowledge of how year-round factors influence population growth, and the demographic mechanisms through which they act, will vastly improve our ability to predict species responses to environmental change and develop effective conservation strategies for migratory animals.

[1] Department of Integrative Biology, University of Guelph, Guelph, Ontario, Canada N1G 2W1. [2] Department of Biology, Bowdoin College, Brunswick, Maine 04011, USA. [3] Swiss Ornithological Institute, Sempach CH-6204, Switzerland. Correspondence and requests for materials should be addressed to B.K.W. (email: bwoodwor@uoguelph.ca) or to D.R.N. (email: rnorris@uoguelph.ca).

Understanding the factors that limit and regulate wildlife populations requires insight into interactions among demographic and environmental processes throughout the annual cycle[1,2]. For migratory species, quantifying seasonal effects of density-dependent and density-independent factors on population dynamics is particularly challenging due to the difficulty of following individuals between their distinct and often distant breeding and non-breeding grounds[3,4]. In birds, variation in the non-breeding environment has been shown to influence individual condition[5], migration timing[6,7] and subsequent reproductive success[6], as well as some components of population growth rate, such as survival and fecundity[8–11]. However, the extent to which density-dependent and -independent effects during the non-breeding period scale up from the individual to affect population growth rate, and the magnitude of these effects relative to those that occur during the breeding period, are unknown. Filling this void requires a full-annual-cycle approach that includes linking the breeding and non-breeding grounds, knowing how density-dependent and -independent factors during different periods of the year influence vital rates, and understanding the relative contribution of those vital rates to population growth rate.

In this paper, we couple multi-year individual tracking data with long-term demographic data for a migratory population of Savannah sparrows (*Passerculus sandwichensis*) breeding on Kent Island (44.48 °N, 66.79 °W) in the Bay of Fundy, New Brunswick, Canada (Fig. 1) to quantify the relative effects of weather and density at the breeding and population-specific wintering grounds on population growth rate via the vital rates. We provide evidence that conditions outside the breeding season limit population growth rate and demonstrate the demographic mechanisms through which limitation occurs.

## Results

**Modelling approach and geolocator results**. To quantify the relative effects of weather and density during the breeding and non-breeding seasons on population growth rate, we first estimated annual vital rates (age- and sex-specific apparent survival, fecundity and sex-specific immigration) and population growth rate. Vital rates and population growth rate were estimated from 26 years of capture-recapture/resighting, population census, and reproductive success data collected on the breeding grounds using an integrated population model (IPM) fitted in a Bayesian framework using Monte Carlo Markov Chain (MCMC) simulations[12]. We then used a novel path analysis approach to estimate (i) direct effects of the different vital rates on population growth rate and (ii) indirect effects of densities (Fig. 2a,b) and weather (daily mean temperature and precipitation; Fig. 2c) at the breeding and population-specific wintering grounds on growth rate via the vital rates. Year-round individual tracking data from light-logging geolocators collected between 2011 and 2014 revealed the population's wintering grounds to be centred in North Carolina (34 °N), at the far eastern extent of the species-wide winter distribution[13], with a range from southern Florida to Pennsylvania[14] (Fig. 1a). To account for uncertainty in vital and growth rate estimates, we fitted the path model to each MCMC sample forming the posterior distribution of the estimated vital rates and population growth rate. Vital rates and effect sizes are presented as means with the upper and lower bounds of the 95% credible interval (CI).

**Demographic contributors to population growth rate**. Population size at the breeding grounds fluctuated between 50 and 114 adults over the course of the 26-year study (Fig. 2a), with population growth rates ranging from 0.52 (95% confidence

interval (CI) = 0.4, 0.56) to 1.67 (95% CI = 1.55, 1.77) (Fig. 3a). Despite equal numbers of years of growth ($\lambda > 1$) and decline ($\lambda < 1$), the population on Kent Island appears to be declining, with current breeding population size only 44% the peak size in 1988 (slope estimate from regressing log-transformed estimated breeding population size versus time = − 0.012 (95% CI = − 0.013, − 0.011)).

As expected, all vital rates (Fig. 3b–e) contributed positively to population growth rate, with the combined effects of adult and juvenile survival contributing most to variation in population growth rate, followed by immigration rate and, lastly, fecundity (Fig. 3f). When we considered effects of age- and sex-specific survival probabilities and sex-specific immigration rates separately, male and female immigration rates contributed most to population growth rate, followed closely by fecundity and adult male survival. Variation in survival of adult females and juveniles of both sexes contributed the least to variation in population growth rate.

**Direct effects of weather and density on vital rates**. Apart from fecundity, all vital rates were strongly correlated with temperatures on the wintering grounds. Annual adult and juvenile survival of both sexes and immigration rate were higher during years of warm winter temperatures (Fig. 4a). Higher survival in warm winters is likely a product of increased food availability relative to cold winters[15,16]. That we also observed a positive correlation between winter temperature and immigration rate supports our assumption that immigrants, many of whom were likely born elsewhere on Kent Island or nearby islands (Fig. 1b), occupied a similar wintering range to Savannah sparrows born on our study site and suggests that weather increased the pool of potential immigrants by increasing overwinter survival. Indeed, years of high survival tended to coincide with years of relatively higher immigration rates for males (correlations between survival and immigration rate: juvenile males = 0.35 (95% CI = − 0.05, 0.67); adult males = 0.37 (95% CI = − 0.01, 0.68)), but less so for females (juvenile females = 0.32 (95% CI = − 0.7, 0.65); adult females = 0.25 (95% CI = − 0.15, 0.59)).

Although non-breeding population density has been found to influence vital rates of other migratory birds during the non-breeding season[11,17,18] and subsequent periods of the annual cycle (for example, sequential density-dependence)[19], none of the vital rates were found to be strongly correlated with wintering Savannah sparrow abundance in our study (Fig. 4a; Supplementary Fig. 1). Population counts from the breeding grounds were consistent with estimates of true population size (Fig. 2a), but post-breeding population size (number of breeding adults plus fledglings) on Kent Island was only weakly correlated with subsequent winter population density (0.06 (95% CI = − 0.08, 0.19)). This suggests either that CBC counts may not have accurately captured interannual variation in winter Savannah sparrow abundance or that other factors along the migration route influence abundance before arrival on the wintering grounds.

In contrast to winter density, population density at the breeding grounds had a greater effect on vital rates compared with weather. Fecundity and adult male survival were both strongly negatively correlated with breeding density (Fig. 4a). Juvenile survival of both sexes was also negatively correlated with density and the strength of density-dependence was similar to the effect of winter temperature (juvenile females: density = − 0.36 (95% CI = − 0.60, − 0.07) versus winter temperature = 0.39 (95% CI = 0.11, 0.61); juvenile males: density = − 0.37 (95% CI = − 0.60, − 0.11) versus winter temperature = 0.35 (95% CI = 0.06, 0.58); Fig. 4a).

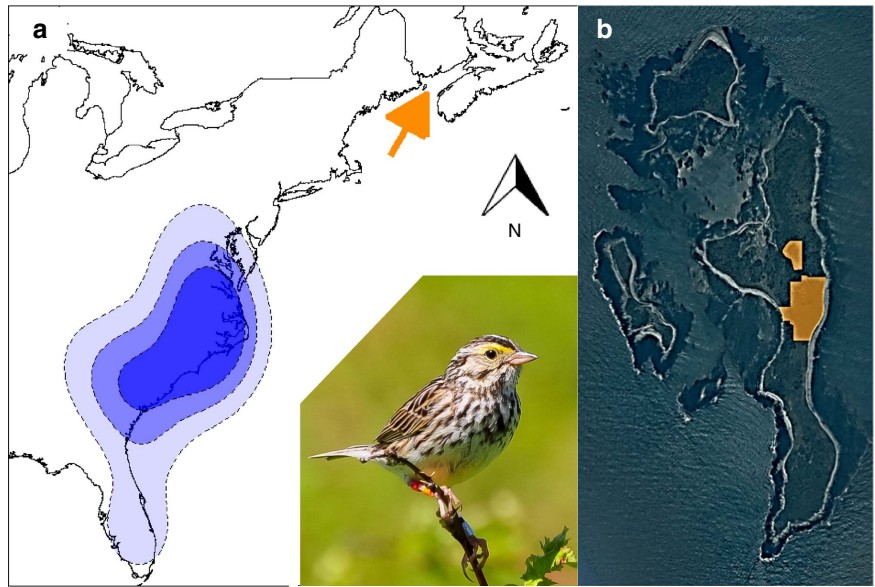

**Figure 1 | Winter distribution and breeding ground study site of Savannah sparrows from Kent Island.** (**a**) Winter distribution (blue contours) of Savannah sparrows from a breeding population on Kent Island, NB, Canada (orange arrow) inferred from geolocator tracking data collected from 2011 to 2014. Blue contours delineate the 30, 50 and 75% kernel densities estimated from the winter locations of 38 individuals. Embedded photo shows a colour-banded Savannah sparrow (photo courtesy of S. Doucet). (**b**) Long-term study area (orange) on Kent Island, with Sheep Island to the west and Hay Island to the north (north arrow in **a** also applies to **b**).

Negative density-dependence of fecundity and survival is likely the result of increased interference competition at high population densities[17,20–22]. In Savannah sparrows, as in most songbirds, males establish and defend breeding territories. At high densities, increased energy expenditure in the defense of territories and paternity[23] may cause adults to be in poorer condition at the end of the breeding season, reducing their chances of survival during subsequent periods of the annual cycle. Similarly, lower fecundity and juvenile survival in years of high breeding density are likely due to increased interference competition leading to smaller territories, reduced food availability, and, in turn, reduced nestling survival and juvenile condition.

If breeding density does indeed influence survival through individual condition, then we might expect effects of subsequent winter temperatures on survival to be stronger after years of high breeding density. However, we found no evidence for an interactive effect of breeding density and winter temperature on annual survival for adults and juveniles of either sex (Supplementary Fig. 2), suggesting that carry-over effects of breeding density on survival through individual condition are stronger during other periods of the annual cycle, such as the post-breeding season or fall migration. Alternatively, breeding density could influence annual apparent survival of juveniles though emigration rather than mortality. If high breeding densities in year $t$ are followed by above average numbers of surviving adults in year $t + 1$, then more first time breeders may be excluded from the breeding population in year $t + 1$. Indeed, breeding density was positively correlated with the number of surviving adults the following year (0.56 (95% CI = 0.37, 0.73)). However, numbers of local recruits in year $t$ and surviving adults in year $t$ were also positively correlated (0.38 (95% CI = 0.08, 0.64)), suggesting that high density of experienced breeders does not necessarily lead to increased exclusion of first time breeders.

In addition to being negatively density-dependent, adult male survival was positively correlated with temperature on the breeding grounds during the pre-breeding season (mid-April to late May). The lack of a similar effect of pre-breeding temperature on female survival is consistent with differences in arrival time between sexes[14]. Males return to the breeding grounds, on average, two weeks earlier than females during a time when weather can be severe (for example, overnight temperatures falling below 0 °C) and food abundance lower compared with when females return. Thus, although arriving early to the breeding grounds can increase reproductive success[14], arriving in harsh springs may reduce survival.

**Indirect effects on population growth rate**. Combining direct effects of the vital rates on population growth rate with direct effects of weather and density at the breeding and wintering grounds on the vital rates, we found clear support for opposing forces of winter temperature and breeding density-dependence driving the dynamics of this migratory population (Fig. 4b). Above-average temperatures on the wintering grounds lead to increased population growth, primarily through increased survival. However, long-term growth of the population was regulated by strong negative density-dependence at the breeding grounds acting to suppress fecundity and survival of adult males and juveniles (Fig. 4b).

## Discussion

A growing body of evidence suggests that non-breeding conditions can have a significant impact on individual success within and across seasons in migratory animals[5–7]. Our results reveal that variation in the non-breeding environment can also scale up to limit population growth rate and, in doing so, strengthen recent calls to approach the study of migratory species from a full-annual-cycle perspective[1,2,4]. We are currently amid a new era of tracking long-distance migration due to the development of miniaturized tracking devices[24]. Major advances in our understanding of the ecology and evolution of migratory species will hinge on how these movement data are integrated with demographic data from both the breeding and non-breeding grounds. Our approach of coupling integrated population modelling with path analysis is an intuitive means by which to

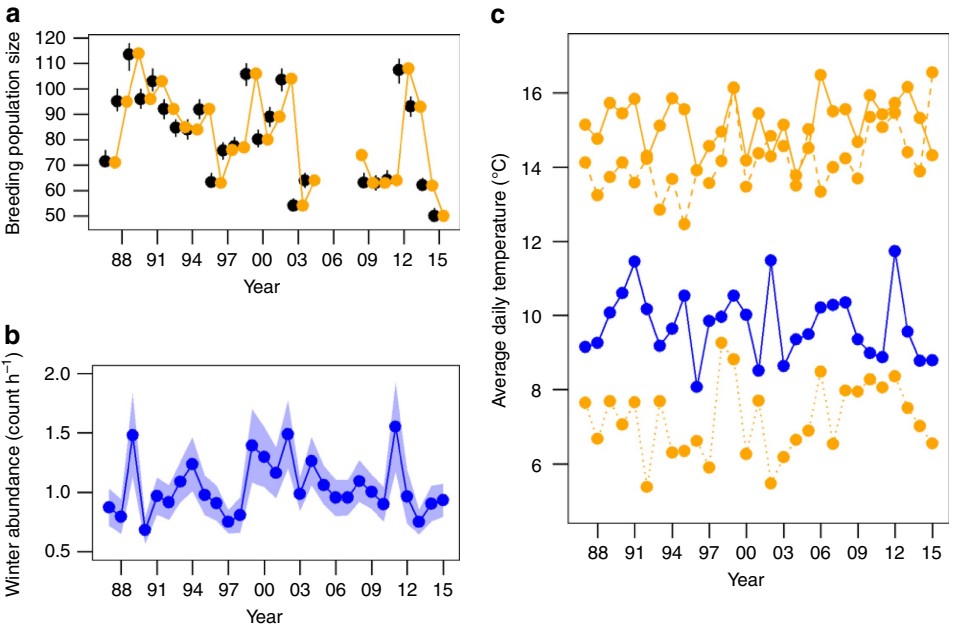

**Figure 2 | Average daily temperatures and Savannah sparrow densities at breeding and wintering grounds.** (**a**) Mean (±95% credible interval) estimated breeding population size (black points) and population counts (orange) at the breeding grounds. (**b**) Mean (±s.e.) number of Savannah sparrows counted per observer hour at Christmas Bird Count survey routes on the wintering grounds. (**c**) Average daily temperatures (°C) at the wintering grounds (blue) and breeding grounds (orange) during the pre-breeding (dotted line), breeding (solid line), and post-breeding (dashed line) seasons.

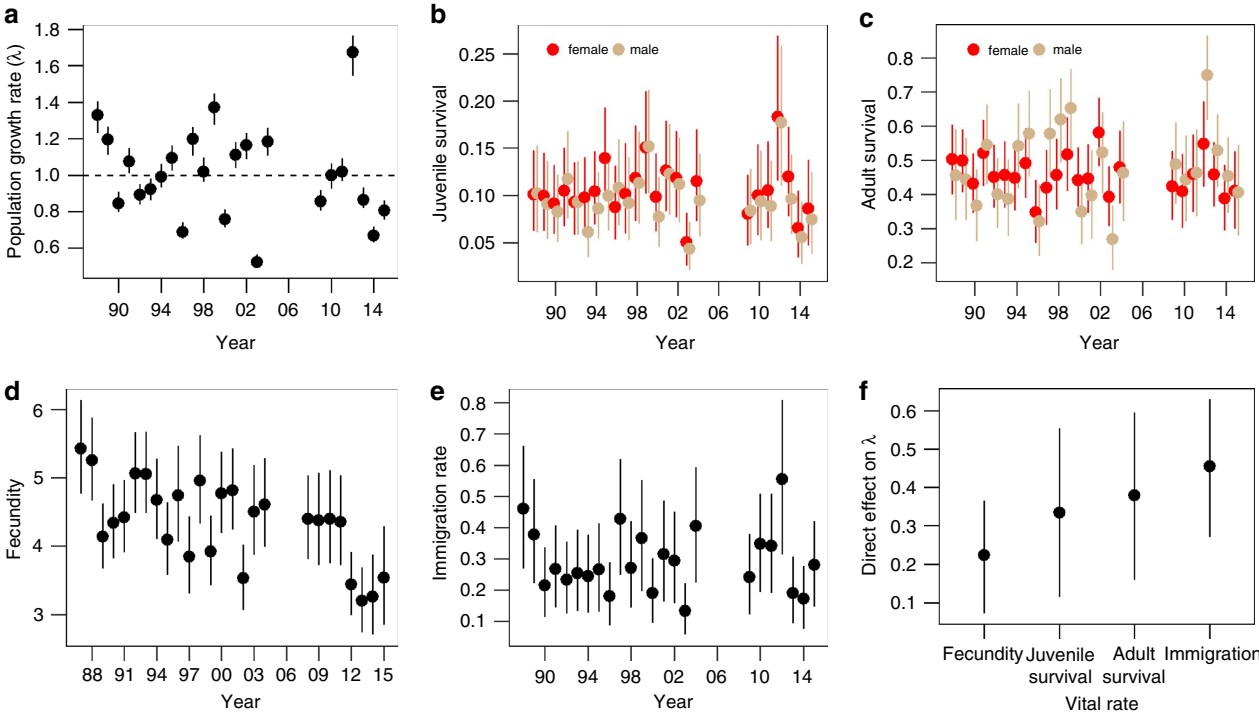

**Figure 3 | Vital rates and their contributions to variation in population growth rate.** Annual estimates (mean ± 95% credible interval) of (**a**) population growth rate, sex-specific (**b**) juvenile and (**c**) adult survival, (**d**) fecundity and (**e**) sex-specific immigration rates for Savannah sparrows from a breeding population on Kent Island, NB, Canada (Fig. 1b). (**f**) Direct effects of variation in juvenile survival, adult survival, fecundity and immigration rate on population growth rate.

quantify the contributions of density-dependent and density-independent factors throughout the annual cycle on population growth rate, while explicitly incorporating uncertainty in parameter estimation.

By accounting for the relative effects of each vital rate on population growth rate, our approach filters out factors that may have a strong direct effect on a given vital rate but that ultimately play a minor role in driving annual variation in population growth rate. Considering the relative contributions of vital rates to population growth rate is important not only for understanding species' population dynamics, but can also help to improve the efficiency of conservation efforts by providing the

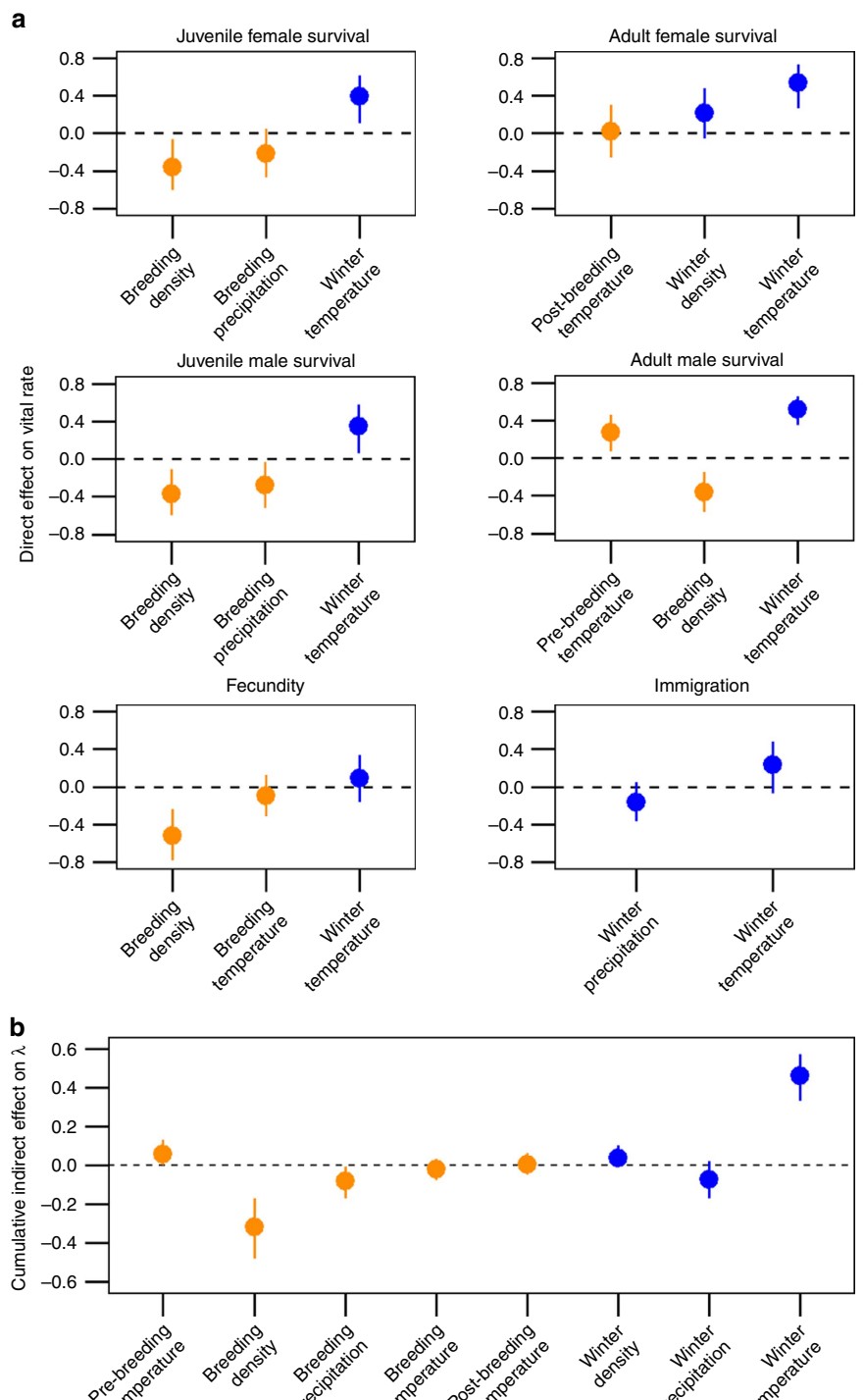

**Figure 4 | Effects of weather and density at breeding and population-specific wintering grounds on population growth rate ($\lambda$).** (**a**) Direct effects (mean ± 95% credible interval) of weather and density at the breeding (orange) and population-specific wintering grounds (blue) on vital rates. (**b**) Cumulative indirect effects of breeding and winter covariates on population growth rate via the vital rates. The total effect of a given covariate $X$ (for example, winter temperature) on population growth rate was calculated from standardized regression coefficients estimated using path analysis by calculating the indirect effects of covariate $X$ on growth rate via each vital rate and then summing indirect effects across all vital rates. The indirect effect of covariate $X$ on growth rate through a given vital rate $V$ equals the product of the direct effect of vital rate $V$ on growth rate and the direct effect of covariate $X$ on vital rate $V$.

components of populations, or periods of the annual cycle, on which targeted conservation actions are likely to have the greatest effect in sustaining populations. For example, warming temperatures are known to influence some migratory animals by altering their timing of breeding in relation to peak food supply for

provisioning young[25]. If fecundity contributes strongly to variation in population growth rate, then the fate of a population or species may hinge on its ability to cope with, or adjust to, long-term warming trends and conservation resources should be allocated to ensure successful reproduction[26].

In contrast, if fecundity contributes relatively little to population growth rate relative to survival, then conservation efforts will be more impactful during the period(s) of the annual cycle that have the greatest influence on survival.

Although our study focused on a single population of a broadly distributed songbird, variation in the non-breeding environment likely plays a primary role in limiting population growth rates of a wide range of migratory birds[20,27], given that non-breeding grounds are typically used during resource-poor periods of the year and that survival is a key vital rate driving population growth in birds[28]. That males and females had similar survival probabilities even though temperatures were, on average, 5 °C higher at the female- versus male-specific wintering grounds, suggests that temperature was not the only environmental factor influencing this vital rate. Other factors, such as differences in landscape cover on the wintering grounds, may also be contributing to variation in annual survival. Also, lower survival of females during other periods of the annual cycle, such as migration or the post-breeding, pre-migration period, relative to males could cause annual survival estimates of the two sexes to balance out.

Whereas density-dependent processes have been shown to regulate other bird populations within and across seasons[11,17–19,29,30], we only found clear evidence for breeding density-dependence regulating population growth. The fact that our study was of a highly philopatric island population could have influenced the strength of density-dependence we observed at the breeding grounds. For example, density-dependence might be stronger in island populations if dispersal-limitation forces individuals to breed at high densities rather than disperse to areas of lower density where resource competition is relaxed. Further, lower species diversity on islands could lead to differences in the strength of density-dependence between island and mainland populations through reduced interspecific competition. Differences in dispersal rates could also explain why we observed density-dependence on the breeding grounds but not at the wintering grounds. During the breeding season, territoriality confines individuals to relatively small areas, leading to frequent aggressive interactions with neighbors. In contrast, during the winter, Savannah sparrows are generally not territorial[13] and thus, have greater flexibility to disperse away from areas of high density.

Finally, it is important to recognize that single breeding populations typically share their non-breeding ranges with individuals from other breeding populations and vice versa[31]. Formation of such migratory networks can lead to dynamics being linked across breeding and non-breeding populations[32]. Therefore, decomposing drivers of dynamics within migratory networks will require information on movements of individuals between breeding and non-breeding grounds as well as between populations. Such a full-annual-cycle, multi-population approach will be critical for predicting how species respond to environmental changes and stressors across their entire ranges[2].

## Methods

**Study population and field methods.** Savannah sparrows (Fig. 1a) are migratory songbirds that occupy all-purpose territories in grasslands and other open habitats across Canada and the northern USA during the breeding season, and form loose flocks across the southern USA and Mexico, as well as parts of Central America during the winter[13,33]. Although still abundant, Breeding Bird Survey data suggest that Savannah sparrow populations are declining[34,35]. The causes of these declines are unknown, but may be related to intensification of agricultural practices, urbanization, and forest regeneration in parts of their range[13,36].

The population of Savannah sparrows we studied breeds in a 10-ha study area on Kent Island (44.48 °N, 66.79 °W) and has been monitored annually since 1987, except for a three-year period from 2005 to 2007. Annual monitoring of the population takes place between late May and the end of July and involves capturing

and colour-banding new members of the study population, re-sighting returning colour-banded individuals, mapping breeding territories, finding nests, and banding nestlings. All capturing of adults and independent juveniles is done using mist-nets. Individuals entering the study population whose natal origins are unknown (immigrants) are marked with a USFWS/CWS aluminum leg-band and a unique random combination of three colour leg-bands. Nestlings born in the study population are first banded in the nest on the seventh day after hatching with a USFWS/CWS aluminum leg-band and one colour leg-band. Those that return to the study population in a subsequent year (local recruits) are re-captured and given a complete three-colour leg-band combination. The sex of immigrants and local recruits is determined at the time of capture by the presence of a cloacal protuberance (males) or brood patch (females). The age of local recruits is known to the day, whereas immigrants are only known to be one year or older.

Breeding territories are determined from daily observations of individual behaviours (for example, locations of singing males) and movements. Savannah sparrows on Kent Island typically form socially monogamous breeding pairs, but a variable fraction of males (15–43%) will secure more than one female[37]. Nests within each territory are typically found at the beginning of incubation and are checked every second day to determine fledging or failure. Incubation lasts an average of 12 days and the nestlings remain in the nest for 9–11 days[38]. On average ( ± s.e.), in each year, 91 ± 2% of females successfully fledge at least one nest (range = 69–100%) and 26 ± 2% (range = 12–48%) successfully fledge a second clutch. Both sexes feed young whereas only females build nests and incubate eggs[13].

Collectively, these monitoring activities have produced 26 years of capture-recapture/resighting (CR), reproductive success, and count data for the population.

All applicable institutional and federal regulations and guidelines for the care and use of animals were followed throughout the execution of this study.

**Geolocators.** Geolocators are archival, battery-powered, animal-borne devices that measure light intensity over time. When recovered from an animal, downloaded light records can be used to determine day length and times of solar noon and midnight, which, in turn, can be used to estimate latitude and longitude for every 12-h period[39]. To determine the primary wintering grounds of our study population, we deployed 163 geolocators ($n_{adults} = 111$ [68%]; $n_{juveniles} = 52$ [32%]) on 148 individuals in Aug/Sep over the course of three years (2011–2013). From 2012 to 2014, we recovered 43 geolocators from 39 different individuals and successfully downloaded data from 42 geolocators representing 38 different individuals. Including individuals that lost their geolocators ($n_{adults} = 13$; $n_{juveniles} = 4$), return rates of adults (0.42) and juveniles (0.25) were similar to estimated age-specific apparent survival probabilities (Fig. 3b,c), suggesting little to no effects of geolocators on survival. Additional methods pertaining to geolocator deployments and position estimation from light data are detailed in ref. 14. Given that geolocator data were collected from 2012 to 2014 (three of the most recent years of the long-term study), our analyses assume that the winter distribution of the study population has remained constant over time.

**Weather data and population density estimates.** Average daily temperature (°C), daily precipitation (mm), and population density estimates were compiled for both the breeding and geolocator-derived wintering grounds of the study population.

Weather at the breeding grounds were obtained from an Environment Canada weather station (www.climate.weather.gc.ca) located ~110 km northeast of Kent Island (45.32 °N, 65.89 °W) and annual breeding population densities were extracted from the long-term demographic data for the study population.

We defined the wintering grounds of the study population as the area within the 50% kernel density contour of the geolocator-derived winter locations of the study population (Fig. 1a). From inside this region, we extracted daily mean temperature and precipitation measurements from 50 airport weather stations using the R package weatherData[40] and Savannah sparrow counts from 127 Christmas Bird Count[41] (CBC; www.christmasbirdcount.org) routes for the period of 1986–2015. We also extracted and summarized weather data for the sex-specific wintering grounds (males wintered on average 275 km farther north than females[14]). Although temperatures were on average nearly 5 °C warmer at the wintering grounds of females than males, temperatures in the two regions were highly correlated ($r = 0.90$) and preliminary analyses revealed little to no difference in effects of temperature on survival when temperature was extracted from sex-specific versus population-wide wintering grounds. Thus, all analyses presented herein use weather and density from the population-wide wintering grounds (Fig. 1a).

After extracting weather and CBC data from the wintering grounds, we summarized winter weather by averaging daily mean temperature and precipitation measurements between 01 Nov and 31 Mar across all weather stations in each year (Fig. 2c). Although weather at specific and shorter time intervals within a season might be expected to have strong effects on vital rates, preliminary analyses using average daily temperature and precipitation summarized at monthly intervals resulted in similar or weaker effects compared with when weather conditions were averaged across the entire winter period. To summarize Savannah sparrow abundance during winter, accounting for differences in survey effort among years, we summed the number of Savannah sparrows counted across all CBC routes and divided by the total number of survey hours. The mean number ( ± s.e.) of

Savannah sparrows observed per survey hour was $0.84 \pm 0.03$ (range $= 0.59$–$1.29$; Fig. 2b). Unlike breeding population size, abundance of Savannah sparrows on the wintering grounds did not show evidence for a temporal trend in either direction (slope estimate $\pm$ s.e. from regressing winter abundance (count h$^{-1}$) versus time $= 0.001 \pm 0.004$, $R^2 = 0.005$), which might be because the population consists of individuals from different breeding populations that may be stable, increasing or decreasing[35].

**Integrated population model.** We used a modified version of the integrated population model (IPM) developed by Schaub et al.[12] to estimate vital rates, population structure and population growth rate from the population count, CR, and reproductive success data (Fig. 3a–e; Supplementary Fig. 3).

At the core of the IPM was a state-space model that described the likelihood of the population count data. The state process was represented by a pre-breeding population projection model that considered three stages and two sexes (hereafter, s denotes sex and can take values of f $=$ female and m $=$ male). The three stages were local recruits R (1-year olds born in the study population the previous year), surviving adults S ($\geq$2-year-old individuals that bred in the study area the previous year), and immigrants I. Immigrants were not known to have bred or been born in the study area the previous year and were assumed to be 1-year olds[42]. For modelling effects of winter weather and density on immigration rates, we further assumed that immigrants shared a similar wintering range as individuals born on Kent Island. This latter assumption was supported by the similar stable-hydrogen isotope content of feathers collected from local recruits (mean $\pm$ s.e. of $-48.0 \pm 1.6$) and immigrants (mean $\pm$ s.e. $= -50.5 \pm 1.5$; see Woodworth et al.[14] for more information on stable isotope analyses).

To account for demographic stochasticity, we projected stage-specific abundances as binomial and Poisson processes. Sex-specific numbers of local recruits in year $t+1$ were projected as $R_{s,t+1} \sim \text{Binomial}(F_{s,t}, \varphi_{j,s,t})$, where $F_{s,t}$ is the sex-specific number of fledglings (F) in year $t$ and $\varphi_{j,s,t}$ is their sex-specific apparent survival probability ($\varphi_j$) from year $t$ to $t+1$. Based on an assumed 50:50 fledgling sex ratio[43], number of female fledglings in year $t$ was projected as $F_{f,t} \sim \text{Binomial}(F_t, 0.5)$ and number of male fledglings in year $t$ was calculated as $F_{m,t} = F_t - F_{f,t}$, where $F_t$ denotes total number of fledglings in year $t$. $F_t$ was projected as a Poisson process, $F_t \sim \text{Poisson}(\rho_t \cdot B_{f,t})$, where $\rho_t$ is fecundity (the number of fledglings produced in year $t$ per female in year $t$) in year $t$ and $B_{f,t}$ is the number of breeding females in year $t$. Sex-specific numbers of surviving adults in year $t+1$ was projected as $S_{s,t+1} \sim \text{Binomial}(B_{s,t}, \varphi_{\text{ad},s,t})$, where $B_{s,t}$ is the sex-specific number of breeding adults in year $t$ and $\varphi_{\text{ad},s,t}$ is sex-specific adult apparent survival probability ($\varphi_{\text{ad}}$) from year $t$ to $t+1$. Lastly, sex-specific numbers of immigrants was projected as $I_{s,t+1} \sim \text{Poisson}(\iota_{s,t+1})$, where $\iota_{s,t+1}$ is the sex-specific expected number of immigrants in year $t+1$. The model assumed that no unmated individuals existed in the population and that individuals reached reproductive maturity in their first year[13]. Following model fitting, population growth rate $\lambda_t$ from year $t$ to $t+1$ was calculated as $\lambda_t = (B_{f,t+1} + B_{m,t+1})/(B_{f,t} + B_{m,t})$, where $B_{s,t} = R_{s,t} + S_{s,t} + I_{s,t}$.

The observation process of the state-space model related sex-specific population counts $C_{s,t}$ to the true sex-specific population sizes $B_{s,t}$ and was modelled with a log-normal distribution and equal variances for the two sexes, $\log(C_{s,t}) \sim \text{Normal}(\log(B_{s,t}), \sigma_t)$. Sex-specific population counts were taken each year at the time of peak breeding activity (median Julian date $= 166$; range $= 156$–$176$) and consisted almost exclusively of colour-banded individuals. Accurate counts of males and females are attainable in this population because unmarked individuals are infrequent, nearly every nest attempted within the study area is found, and sexes are distinguishable based on behaviour (for example, males singing and females incubating). That nearly every counted individual is colour-banded means that there was considerable overlap of the population count and mark-recapture data sets. Although estimation of the joint likelihood of the IPM assumes independence of the different data sets, simulation studies have revealed that estimates of demographic parameters, including those for which no data are available, are robust to violations of independence[44,45]. The state-space model was initialized using priors for the stage-specific abundances in the first year (see Supplementary Information for model code).

CR data were used to estimate apparent survival and recapture/resighting probabilities ($p$) and the likelihood of this data set was described using a Cormack-Jolly-Seber (CJS) model[46]. Apparent survival is the probability of an individual surviving from year $t$ to $t+1$ and returning to the study area in year $t+1$ and recapture/resighting is the probability of detecting an individual given that they are alive and present in the study area in year $t+1$. We considered two age classes (adult ad and juvenile j) and both sexes in the estimation of both parameters. Juvenile survival $\varphi_{j,s,t}$ is the probability of a fledgling born in the study area in year $t$ surviving and returning to the study area in year $t+1$, whereas adult survival $\varphi_{\text{ad},s,t}$ is the probability of an individual that bred in the study area in year $t$ surviving and returning to the study area in year $t+1$. For estimation of juvenile survival, we again assumed a 50:50 fledgling sex ratio[43]. Recapture/resighting probabilities were fixed to zero from 2005 to 2007 when population monitoring was interrupted.

Lastly, reproductive success data contributed to the estimation of fecundity ($\rho$) and the likelihood of this data set was described using a Poisson regression model, $J_t \sim \text{Poisson}(\rho_t \cdot C_{f,t})$, where $J_t$ is the number of fledglings produced in year $t$ and $C_{f,t}$ is the number of surveyed females in year $t$.

**Model implementation and goodness-of-fit.** We analysed the IPM in a Bayesian framework using Markov Chain Monte Carlo simulations, which we implemented in JAGS[47] using the R package jagsUI[48]. Vague prior distributions were specified for all parameters (see Supplementary Information for model code). We ran three independent chains with different starting values for 1,000,000 iterations. We used a burn-in of 500,000 iterations and kept every hundredth sample, resulting in 15,000 posterior samples. Convergence of model chains was assessed using Gelman-Rubin $\hat{R}$ diagnostic statistics[49] and was reached for all parameters ($\hat{R} < 1.005$).

Measures of goodness-of-fit for state-space models and IPMs are not well established[50,51]. Therefore, we instead evaluated fit of the CJS and Poisson models to the individual CR and reproductive success data sets[52]. Fit of the CJS model was evaluated using the Freeman-Tukey statistic[53] and fit of the Poisson model was evaluated using the Chi-square discrepancy measure. Both methods involved simulating expected data from the model and then comparing expected to observed data. We found little evidence for lack of fit of the CJS and Poisson models to the CR and reproductive success data sets (Supplementary Fig. 4).

**Temporal variability of vital rates.** To quantify the temporal variability of the vital rates, we modelled each vital rate with a mean $\mu$ and temporal residual $\varepsilon_t$. Following the approach of ref. 54 as implemented in ref. 12, $\varepsilon_t$ were assumed to originate from a multivariate normal distribution with a mean of 0, $\varepsilon_t \sim \text{MVN}(0, \Sigma)$, where $\Sigma$ is the variance-covariance matrix for the vital rates. Age- and sex-specific survival probabilities were modelled using logit link functions, and fecundity and the expected number of immigrants were modelled using log link functions. This parameterization of the vital rates also allowed us to estimate effects of variation in weather and density at different periods of the annual cycle on each:

$$\text{logit}\left(\varphi_{\text{ad},s,t}\right) \sim \mu_{\varphi_{\text{ad},s}} + \beta_{\text{ad},s} \cdot X_t + \varepsilon_{\varphi_{\text{ad},s,t}} \quad (1)$$

$$\text{logit}\left(\varphi_{j,s,t}\right) \sim \mu_{\varphi_{j,s}} + \beta_{j,s} \cdot X_t + \varepsilon_{\varphi_{j,s,t}} \quad (2)$$

$$\log(\rho_t) \sim \mu_\rho + \beta \cdot X_t + \varepsilon_{\rho,t} \quad (3)$$

$$\log(\iota_{s,t}) \sim \mu_{\iota_s} + \beta_s \cdot X_t + \varepsilon_{\iota_{s,t}} \quad (4)$$

In equations (1–4), $X_t$ represents annual values of an explanatory variable of interest (for example, winter temperatures) and $\beta$ denotes the estimated slope coefficient.

Age- and sex-specific recapture/resighting probabilities were also modelled on the logit scale with a mean $\mu$ and temporal residual $\varepsilon_t$, but were assumed to vary independently over time:

$$\text{logit}(p_{\text{age},s,t}) \sim \mu_{p_{\text{age},s}} + \varepsilon_{p_{\text{age},s,t}} \quad \varepsilon_{p_{\text{age},s,t}} \sim N\left(0, \sigma^2_{p_{\text{age},s}}\right) \quad (5)$$

Excluding 2005–2007 when population monitoring was interrupted, mean recapture/resighting probabilities were high for all four age-sex groups (adult males $= 0.96$ (95% CI $= 0.92$, 0.99); adult females $= 0.96$ (95% CI $= 0.93$, 0.98); juvenile males $= 0.93$ (95% CI $= 0.86$, 0.98); juvenile females $= 0.93$ (95% CI $= 0.87$, 0.97)).

**Weather and density variable selection.** We conducted a variable selection procedure to reduce the set of variables considered in the final path model of factors limiting and regulating population growth rate. Using the framework for decomposing temporal variability in the vital rates described in the previous section, we fit a series of models in which each vital rate was written as a linear function of a single weather or population density variable from a given period of the annual cycle. We considered average daily mean temperature and precipitation from 01 Nov to 31 Mar at the wintering grounds and average daily mean temperatures and precipitation for three distinct periods at the breeding grounds: pre-breeding (15 Apr to 19 May), breeding (20 May to 31 Jul), and post-breeding or pre-migration (15 Aug to 30 Sep). For each vital rate, we then selected the two weather variables, one from the breeding grounds (encompassing the pre-breeding, breeding, and post-breeding periods) and one from the wintering grounds, for which the 90% credible interval of the slope coefficient overlapped 0 the least. If the 90% credible interval of the slope coefficient excluded 0 for more than one variable from the breeding or wintering grounds, then all such variables were carried forward to the final path model.

To assess density-dependence at the breeding and wintering grounds, we considered peak population size ($C_{m,t} + C_{f,t}$) and CBC-derived counts corrected for survey effort, respectively. Because both fecundity and population size at the breeding grounds have shown some evidence for a decline over time (Fig. 3d), we regressed fecundity in relation to de-trended population density at the breeding grounds to avoid spurious detections of density dependence[55,56].

All variables were standardized by subtracting the mean and dividing by the standard deviation. Because vital rates estimated from 2005–2007 (the years during which population monitoring did not occur) approximated the long-term mean, effect sizes of weather and density on vital rates were likely underestimated. See Supplementary Figs 1, 5 and 6 for results of weather and density variable selection.

**Direct and indirect effects on population growth rate.** We adopted a novel path analysis[57] approach that used annual estimates of vital rates and population growth rate (Fig. 3a–e) and key weather and density variables (Fig. 2; Supplementary Figs 1,5 and 6) to quantify the relative effects of weather and density at the breeding and wintering grounds on population growth rate via the vital rates. Advantages of this approach include the flexibility and accessibility of using common regression techniques to estimate direct and indirect effects of covariates on response variable(s) and the ability to account for uncertainty in model inputs (for example, vital rates) by fitting the model to each sample of their posterior distributions. The path model consisted of seven normal linear models, one relating population growth rate to the vital rates and the remaining six relating the vital rates to weather and density at different periods of the annual cycle:

$$\lambda_t \sim \varphi_{j,f,t} + \varphi_{ad,f,t} + \varphi_{j,m,t} + \varphi_{ad,m,t} + \rho_t + \omega_t \quad (6)$$

$$\varphi_{j,f,t} \sim \text{breeding density}_{t-1} + \text{breeding precipitation}_{t-1} + \text{winter temperature}_t \quad (7)$$

$$\varphi_{ad,f,t} \sim \text{winter density}_t + \text{postbreeding temperature}_{t-1} + \text{winter temperature}_t \quad (8)$$

$$\varphi_{j,m,t} \sim \text{breeding density}_{t-1} + \text{breeding precipitation}_{t-1} + \text{winter temperature}_t \quad (9)$$

$$\varphi_{ad,m,t} \sim \text{breeding density}_{t-1} + \text{prebreeding temperature}_t + \text{winter temperature}_t \quad (10)$$

$$\rho_t \sim \text{breeding density}_t + \text{breeding temperature}_t + \text{winter temperature}_t \quad (11)$$

$$\omega_t \sim \text{winter temperature}_t + \text{winter precipitation}_t \quad (12)$$

Because population growth rate is more naturally expressed in terms of an immigration rate, we parameterized immigration as a rate in the path model, rather than as a number of immigrants as in the IPM. Furthermore, given that numbers of male and female immigrants were similarly affected by weather at the wintering grounds (Supplementary Fig. 6) and that immigration is the vital rate for which we had the least information (for example, natal origins and wintering locations of immigrants were unknown), we combined estimated numbers of male and female immigrants to derive a single, population-wide annual estimate of immigration rate, where $\omega_t = (I_{f,t+1} + I_{m,t+1})/(B_{f,t} + B_{m,t})$. Because winter density had little to no effect on sex-specific immigration (Supplementary Fig. 1), we excluded winter density from the model of immigration rate.

To account for uncertainty in the estimated vital rates, we fitted the path model to each of the 15,000 posterior samples. We excluded vital rate and population growth rate estimates for the three years (2005–2007) for which demographic data and population counts were not collected. At each iteration of the path model (i) all vital rates, population growth rates, and weather and density variables were scaled by subtracting the mean and dividing by the standard deviation, and (ii) standardized slope coefficient estimates from each of the seven sub-models were used to calculate indirect effects of weather and density on population growth rate via the vital rates. The indirect effect of a given covariate $X$ on population growth rate via a given vital rate $V$ was calculated as the product of the direct effect (standardized regression coefficient) of covariate $X$ on vital rate $V$ and the direct effect of vital rate $V$ on population growth rate. Indirect effects for a given covariate $X$ were then summed across vital rates to determine the cumulative indirect effect of a given covariate on population growth rate.

**Code availability.** R and BUGS code for the integrated population model and path analysis are available in the Supplementary Information file as Supplementary Methods.

**Data availability.** Data are available from the corresponding author (B.K.W.) upon request.

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

## Acknowledgements

We thank the many researchers, students, field assistants, and volunteers that have contributed to the long-term Savannah sparrow study on Kent Island, particularly G.W. Mitchell and R.A. Mauck. Christmas Bird Count data were provided by the National Audubon Society and through the generous efforts of countless volunteers across the western hemisphere. Funding was provided through a Discovery Grant (D.R.N.) and Canada Graduate Scholarship (B.K.W.) from the Natural Sciences and Engineering Research Council of Canada, a University Research Chair (D.R.N.) and Graduate Scholarship (B.K.W.) from the University of Guelph, the Canada Foundation for Innovation (D.R.N.), an American Ornithologist's Union Student Research Award (B.K.W.), a Society of Canadian Ornithologists Taverner Research Award (B.K.W.), a Cooper Ornithological Society Mewaldt-King Student Research Award (B.K.W.), the National Science Foundation (N.T.W.; OPUS award no 0816132), and Bowdoin College (N.T.W.). This represents Bowdoin Scientific Station contribution no. 253.

## Author contributions

B.K.W. and D.R.N. conceived and designed the study. B.K.W., N.T.W., A.E.N. and D.R.N. collected the data; B.K.W. and M.S. analyzed the data with assistance from D.R.N.; B.K.W. and D.R.N. wrote the manuscript; all authors reviewed and edited the manuscript.

## Additional information

**Competing interests:** The authors declare no competing financial interests.

**Publisher's note**: 

