## [Peer Review File · Nature Communications]

Reviewers' Comments:

Reviewer #1 (Remarks to the Author):

This is a wonderful report on a long-term effort on the population of a well-defined island population of a migratory songbirds.

I have no qualms with the structure and methods underlying this paper, and it is very clearly written.

These are the items that would do with some more thought and elaboration:

General: the fact that this is a site-faithful island population may have strengthened density dependent effects: is there evidence for this, would this need a bit of elaboration to generalize the interpretations?

Line 82 – (reduced thermoregulatory costs and increased food abundance relative to cold winters¹⁷): I wonder whether this should be rewritten a bit. Higher thermoregulatory costs are almost never a problem per se as long as there is enough food (e.g. publications by Ruthrauff on the most northerly wintering shorebirds in the world, rock sandpipers) and I would guess that what matters here in the temperature effects is reduced food abundance due to snow cover of freezing over rather than the thermoregulatory costs. Is there a meteorological measure (I guess this is too far south for snow cover) that can capture survival and immigration rates even better?

Lines 137-141: I am puzzled why the otherwise great Discussion is limited here to songbirds. The study has a much greater relevance to all kinds of migrants and I would especially like to see the proposal by Rakhimberdiev et al (in *J Avian Biol* 46, 332-341) on the issue of sequential density dependence (highly relevant in this case) and the expectation that peaks in seasonal mortality (admittedly not specifically addressed here) to be discussed within this illuminating Savannah sparrow framework.

Please clean up the reference section (now still with copied Capitalized titles, usually indicative of a quite careless editorial hand...).

Reviewer #3 (Remarks to the Author):

This MS addresses which density dependent processes and which environmental factors regulate/affect a breeding population of Savannah sparrows, studied for a long period on an island in Canada. The authors use their data on reproduction and mark-recapture data to analyse the contribution of various demographic rates to population growth and how these are being affected by conditions at the breeding grounds, as well as during the wintering grounds. They can develop such a whole annual cycle perspective, because from earlier geolocator work they have a fair idea where individuals from their breeding population winter. Their main result is that they only find density dependent processes during the breeding season, and that winter weather affects survival. Hence this population seems to be mostly limited in size during winter, but most regulation happens during the breeding season.

General remarks

I highly appreciate this long-term study (as I do with all high quality long-term studies!), and the use of the data to analyse some of the basic processes affecting the population dynamics. In most migrant species it is not very well known how conditions during the entire annual cycle affect their dynamics, and the authors have tried to provide such a year-round perspective which I consider as an important (but not essentially new, see below) contribution. The statistical models applied are cutting-edge, but I have to admit that I find these difficult to judge as I am rather unfamiliar with these models, and they are quite briefly described. The paper is furthermore well written.

My general judgement of the MS is a bit mixed: on the one hand I do very much appreciate the focus on a whole annual cycle perspective in analyzing environmental drivers of population dynamics, especially in migrants. On the other hand, I find that so much ecological details are missing, that I am not convinced by one of the key results: that density dependence does not occur during the non-breeding season. I have the following general remarks, which I hope help to improve this MS:

1. One of the key contributions is that the authors can relate site-specific factors of the wintering grounds of a breeding population from their previous geolocator studies. When reading their interesting paper on this, it is striking that they find a difference between the sexes, and indications that juveniles winter further south (isotope work). This seems to be mostly ignored, as the authors take only a single wintering area for all sex/age classes.
2. The winter density being used is not very well substantiated, and we get no idea how good data from these Christmas bird counts are. How many sparrows are being observed on average per route? If one e.g. analyses these data (count/route/year), is there then significant variation between years, or is the variation within routes enormous? If the authors want to support their claim of an absence of density dependence in winter, they have to much better substantiate the value of these estimates. It would also be interesting to know whether these values show a trend over the years. And for a density dependent process, one would expect that

the interaction density * winter conditions would show up, but I think that interactions are not being tested (and may not be possible with the data).

3. For quite some short-distance migratory species it is well known that they can flexibly adjust their wintering location based on encountered wintering conditions, moving further south during harsh winter weather. Is anything known about this for SS? Also: these geolocator data are from recent origin, and evidence is accumulating that especially short distance migrants shift their wintering range northwards. If this would be true, the whole rationale of using current wintering ranges for a long-term study does not really hold true.

4. Although the effects of winter mean temperature are clearly visible, it is biologically not clear whether it is mostly the mean temperature over such a long period that determines survival, or whether it is e.g. a period of harsh conditions. In principal a very mild winter still can have a low survival if there is heavy snow cover for a single week.

5. In general, I am missing a lot of ecological details on the population and the stats, that make the MS difficult to judge. I realise that for a paper in such a journal space constraints are severe, but also in the supplementary material there is rather little info. It is e.g. not clear what (re)capture rates are, whether the models of mark-recapture do substantiate sex*year effects, what the population trend is, whether the local (post-)breeding population and winter density are correlated, whether birds make multiple clutches, etc.

6. I have some reservation in determining density dependence in populations that show a clear trend over time. Density dependence requires a certain carrying capacity (determined by environmental conditions such as food) and a regulating process that makes the population to grow when being at relatively low density and decline when being relatively high. In case of a population trend, carrying capacity is apparently declining, and it becomes not so apparent how density dependence would operate and how it can be estimated. The authors do acknowledge this by using detrended breeding population size, but this may not solve the whole problem. It would be interesting also if the authors do mention whether this population decline is a more global pattern, and whether anything is known what causes this.

7. I find it striking that the authors are not citing one of the best studies on the effects of year-round density on the dynamics of populations, which is of the Icelandic black-tailed godwits by Jenny Gill and co-workers (Gill et al. 2001, Gunnarsson et al. 2005). In their study they knew exactly in which estuary individuals were wintering, and how during population growth sequentially more low quality sites were occupied with lower survival through a density dependent process (which also operates at the breeding grounds).

References

- Gill J.A., Norris, K., Potts, P.M., Gunnarsson, T.G., Atkinson, P.W. & Sutherland, W.J. 2001. The buffer effect and large-scale population regulation in migratory birds. *Nature* 412: 436-438.
- Gunnarsson T.G., Gill, J.A., Petersen, A., Appleton, G.F. & Sutherland, W.J. 2005. A double buffer effect in a migratory shorebird population. *J. Anim. Ecol.* 74: 965-971.

Reviewer #4 (Remarks to the Author):

This manuscript estimates the influence of weather variables and density dependence on demographic parameters of a migratory songbird species. The authors conclude that their study population is regulated by winter temperatures affecting survival probability, but that increased winter survival can lead to higher population density on breeding grounds which in turn reduces survival of males and fecundity. The manuscript is very well written, and the analyses are both innovative and robust and support the conclusions. Because migrants are experiencing global population declines, the formal framework presented here will allow us to objectively quantify the underlying drivers of population declines for other species, which is a much-needed advancement.

For many declining migrant populations the lack of knowledge about the location of their non-breeding range hampers efforts to assess climatic influences on demographic parameters. In this manuscript the authors use tracking data of birds from their study population to delineate an area over which they obtain winter weather data to ensure that influences of winter weather on demographic parameters are measured in the appropriate region. This approach, and the integrated population model used to estimate demographic parameters, are robust and convincing.

I found only three issues that would benefit from some clarification:

- 1) The population count data, which are a critical component of the IPM, are not very well explained. The authors mention territory mapping (L. 155), and that the species is mostly monogamous (L. 166-168), but I think it would be important to specify how each component of the population is counted. In many songbirds we often just count the (singing) males, and simply assume that there are as many females. From the model code it is evident that there are count data for breeding males and breeding females, and it would be good to provide a better description of how the count data were obtained, especially given that some males may 'secure more than one female' (L. 168).
- 2) An integrated population model works best if the different data sources (count data, mark-recapture data) are independent, but the models are fairly robust even if there is substantial overlap in the data sets (Abadi et al. 2010). From the description of the territory mapping effort it

appears likely that the population census data are comprised of counts of color-banded birds, hence the count data may contain exactly the same individuals as the capture-resight data used for the estimation of survival. Although the paper by Abadi et al. (2010) demonstrated that IPMs are robust to violations of the independence assumption, these simulations did not examine estimation of parameters for which no data existed (e.g. immigration), and it would be good to state the extent of the overlap of the two data sets (did unmarked individuals contribute to the population count data?), and how this partial (or complete) non-independence may affect demographic parameter estimates.

3) I am not familiar with the path analysis, and cannot judge how robust this approach is compared to standard model selection approaches, which are arguably difficult for complex Bayesian models such as this IPM. Because the path analysis is a fairly innovative component, it would be good to briefly highlight the main advantages of this approach over e.g. Gibbs variable selection or the product space method [which may be rather awkward to implement in a complex IPM] - see Tenan et al. 2014. *Ecol. Modell.* 283:62-69.

Minor comments:

L. 87-90: This statement is a bit tenuous for females, because the credible intervals of the correlation overlap 0 quite substantially.

L. 179: Although irrelevant for the purpose of this manuscript, there is generally huge concern over the effects of tags on birds. Unless GLS were deployed on juveniles this recovery rate (26%) is a bit lower than the average adult survival probability (~40%, Fig. 2). It would be good to add a comment how you might explain that discrepancy.

L. 227: It might be good to move L. 260 to here to clarify that $B_{s,t} = R_{s,t} + S_{s,t} + I_t$. Based on the text up to here the recruits and immigrants don't really enter the breeding population, which I found initially confusing (although it is evident from the model code and the equation in L. 260 that immigrants and recruits enter the breeding population).

L. 259: Why did you not include this calculation as a derived parameter in the model to ensure adequate error propagation? It is obviously just as valid to use the `sims.list` output to do that - I'm just curious why it was done outside the model. Were there computational benefits?

R-code: Thank you for including the code, which is fantastic to properly understand the model!

Reviewers' comments:

Reviewer #1 (Remarks to the Author):

This is a wonderful report on a long-term effort on the population of a well-defined island population of a migratory songbirds.

I have no qualms with the structure and methods underlying this paper, and it is very clearly written.

Author response: We thank the reviewer for their constructive and positive assessment of our manuscript.

These are the items that would do with some more thought and elaboration:

General: the fact that this is a site-faithful island population may have strengthened density dependent effects: is there evidence for this, would this need a bit of elaboration to generalize the interpretations?

Author response: We agree with the reviewer that the potential exists for strength of density-dependence to vary between highly philopatric island populations and mainland populations and we have included discussion of this in the text (lines 161-175). One case in which density-dependence might be stronger in island populations is if dispersal-limitation forces island populations to breed at high densities rather than disperse to areas of lower density where competition for resources is lower and, in turn, reproductive rates potentially higher. Alternatively, if island habitats are of higher quality compared to mainland habitats, then individuals may be better served to breed at high densities and not disperse, in which case density-dependent effects may be weaker on islands. Although high density does not always indicate high habitat quality, breeding densities of Savannah sparrows tend to be highest on islands (Wheelwright & Rising 2008) suggesting that island habitats might be of higher quality (e.g., fewer mammalian predators and competing species). Island populations have also likely adapted/adjusted to breeding at high densities, such that strength of density-dependence may be similar in populations occupying islands vs. more contiguous mainland habitats.

Reference:

Wheelwright, N. T. & Rising, J. D. Savannah sparrow (*Passerculus sandwichensis*). *Birds N. Am. Online* <http://bna.birds.cornell.edu/bna/species/045/articles/introduction> (2008).

Line 82 – (reduced thermoregulatory costs and increased food abundance relative to cold winters¹⁷): I wonder whether this should be rewritten a bit. Higher thermoregulatory costs are almost never a problem per se as long as there is enough food (e.g. publications by Ruthrauff on the most northerly wintering shorebirds in the world, rock sandpipers) and I would guess that what matters here in the temperature effects is reduced food abundance due to snow cover of freezing over rather than the thermoregulatory costs.

Author response: We thank the reviewer for highlighting the studies of Ruthrauff and colleagues. As suggested, we have adjusted our discussion of the effect of temperature on survival and have included a reference to Ruthrauff et al. (2013) to support the interpretation that the main effect of low temperatures is reduced food abundance and that thermoregulatory costs are less likely to be important except in periods of extreme cold (lines 81-82).

Reference:

Ruthrauff, D. R., Gill Jr, R. E. & Tibbitts, T. L. Coping with the cold: an ecological context for the abundance and distribution of Rock Sandpipers during winter in upper Cook Inlet, Alaska. *Arctic* 269–278 (2013).

Is there a meteorological measure (I guess this is too far south for snow cover) that can capture survival and immigration rates even better?

Author response: We agree with the reviewer that other weather variables or the same variables evaluated at different temporal scales could potentially capture variation in survival and immigration better than average temperature and precipitation values across the full winter period. For this reason, we conducted preliminary

analyses where we averaged daily temperature and precipitation at monthly intervals as well as over the entire winter period. We also considered the proportion of days with temperatures below 0°C at monthly intervals and across the entire winter period. To summarize, for all four age-sex groups, average daily temperature over the entire winter had a greater (or equally strong) effect on survival compared to average daily temperature in any one month or proportion of days with temperatures below 0°C. Results for precipitation were more variable across age-sex groups, but effects were generally weak (the only monthly effect whose 90% credible interval did not overlap zero was for adult female survival), so we opted to average precipitation across the entire winter period for consistency and simplicity. There are certainly even more winter weather variables (or the same variables could be explored at even finer temporal scales, e.g., weekly or bi-weekly), but the expected change (if any) would be for an increase in what are already clearly limiting effects of winter weather. Thus, any such additional analyses are highly unlikely to change the main results of the study. In the revised manuscript, we acknowledge having conducted these preliminary analyses on lines 247-253, 256-260. We have decided not to include these in the Supplementary Information simply because there were several scenarios analyzed and they are quite lengthy.

Lines 137-141: I am puzzled why the otherwise great Discussion is limited here to songbirds. The study has a much greater relevance to all kinds of migrants and I would especially like to see the proposal by Rakhimberdiev et al (in *J Avian Biol* 46, 332-341) on the issue of sequential density dependence (highly relevant in this case) and the expectation that peaks in seasonal mortality (admittedly not specifically addressed here) to be discussed within this illuminating Savannah sparrow framework.

Author response: As suggested, we have adjusted discussion of our results beyond songbirds in the revised manuscript (lines 146-156, 161-175), including several citations of non-songbird studies (lines 82, 92-93, 161-163). We also reference the possibility for sequential density-dependence to play a regulating role in this and other populations.

Please clean up the reference section (now still with copied Capitalized titles, usually indicative of a quite careless editorial hand...).

Author response: As suggested, the reference section has been tidied up (lines 436-571).

Reviewer #3 (Remarks to the Author):

This MS addresses which density dependent processes and which environmental factors regulate/affect a breeding population of Savannah sparrows, studied for a long period on an island in Canada. The authors use their data on reproduction and mark-recapture data to analyse the contribution of various demographic rates to population growth and how these are being affected by conditions at the breeding grounds, as well as during the wintering grounds. They can develop such a whole annual cycle perspective, because from earlier geologist work they have a fair idea where individuals from their breeding population winter. Their main result is that they only find density dependent processes during the breeding season, and that winter weather affects survival. Hence this population seems to be mostly limited in size during winter, but most regulation happens during the breeding season.

General remarks

I highly appreciate this long-term study (as I do with all high quality long-term studies!), and the use of the data to analyse some of the basic processes affecting the population dynamics. In most migrant species it is not very well known how conditions during the entire annual cycle affect their dynamics, and the authors have tried to provide such a year-round perspective which I consider as an important (but not essentially new, see below) contribution. The statistical models applied are cutting-edge, but I have to admit that I find these difficult to judge as I am rather unfamiliar with these models, and they are quite briefly described. The paper is furthermore well written.

My general judgement of the MS is a bit mixed: on the one hand I do very much appreciate the focus on a whole annual cycle perspective in analyzing environmental drivers of population dynamics, especially in migrants. On the other hand, I find that so much ecological details are missing, that I am not convinced by one of the key results: that density dependence does not occur during the non-breeding season. I have the following general remarks, which I hope help to improve this MS:

Author response: We thank the reviewer for their positive comments and constructive feedback on our manuscript. As will become apparent from our responses below, we have added considerable additional ecological, methodological, and statistical details to the revised manuscript, including those pertaining to our assessment of non-breeding density-dependence, which we hope will alleviate their concerns.

1. One of the key contributions is that the authors can relate site-specific factors of the wintering grounds of a breeding population from their previous geolocator studies. When reading their interesting paper on this, it is striking that they find a difference between the sexes, and indications that juveniles winter further south (isotope work). This seems to be mostly ignored, as the authors take only a single wintering area for all sex/age classes.

Author response: As the reviewer suggests, a previous study of Savannah sparrows from Kent Island revealed that males overwintered ~275 km farther north than females, which could result in males and females experiencing different conditions during winter. For this reason, we conducted preliminary analyses to evaluate effects of average daily temperature (evaluated at monthly intervals) from the population-wide and sex-specific wintering grounds on survival. These analyses revealed little to no difference in the effect of temperature on survival when extracted from the population-wide vs. sex-specific wintering grounds. Although temperatures were on average almost 5°C warmer on the wintering grounds of females compared to males, temperatures at the sex-specific wintering grounds were highly correlated ($r = 0.90$). In the revised manuscript, we acknowledge having conducted these preliminary analyses (lines 247-253). We have not included them in the Supplementary Information simply because there were several scenarios analyzed and they are quite lengthy.

2. The winter density being used is not very well substantiated, and we get no idea how good data from these Christmas bird counts are. How many sparrows are being observed on average per route? If one e.g. analyses these data (count/route/year), is there then significant variation between years, or is the variation within routes enormous? If the authors want to support their claim of an absence of density dependence in winter, they have to much better substantiate the value of these estimates.

Author response: As suggested, we have provided more information on our use of Christmas Bird Count data in the revised manuscript (lines 95-100, 263-268, Fig. 2b). Importantly, we have adjusted the reporting of our results and discussion pertaining to effects of winter density to reflect that the lack of evidence we found for effects of winter density on the vital rates should not be interpreted as definitive evidence against winter density-dependence. This is because although interannual variation in average numbers of Savannah sparrows counted per survey hour existed, within-year variation across routes was substantial. Moreover, post-breeding population size was only weakly correlated with winter population density.

It would also be interesting to know whether these values show a trend over the years.

Author response: As suggested, we explored temporal trends in CBC counts of Savannah sparrows and found that they neither increased nor declined over time. This information has been added to the revised manuscript (lines 265-266).

And for a density dependent process, one would expect that the interaction density * winter conditions would show up, but I think that interactions are not being tested (and may not be possible with the data).

Author response: We agree with the reviewer that effects of winter conditions and density may not be independent of one another. However, given the weak main effects of winter density on vital rates and the uncertainty in winter density estimates, we do not feel that further extending our models to estimate an interaction effects would improve our inferences regarding non-breeding population limitation and regulation in this situation.

3. For quite some short-distance migratory species it is well known that they can flexibly adjust their wintering location based on encountered wintering conditions, moving further south during harsh winter weather. Is anything known about this for SS?

Author response: There are no studies that have examined inter-annual variation in winter latitude of Savannah sparrows and whether such variation may be linked to weather. From our three years of geolocator data, we found some evidence of interannual variation in winter latitude (see figure lower-left), but the difference was not significant after controlling for age and sex (see Table 1 in Woodworth et al. 2016). Average winter latitude was most northerly in 2012, which corresponded to the warmest winter not only over the three-year period geolocators were deployed (see figure lower-right; blue lines denote years in which geolocators were deployed) but the entirety of the long-term study. However, a proper assessment of the link between winter conditions and inter-annual variation in winter latitude, as well as the implications for population dynamics, will require additional years of tracking data.

Reference:

Woodworth, B. K. *et al.* Differential migration and the link between winter latitude, timing of migration, and breeding in a songbird. *Oecologia* 181:413–422 (2016).

Also: these geolocator data are from recent origin, and evidence is accumulating that especially short distance migrants shift their wintering range northwards. If this would be true, the whole rationale of using current wintering ranges for a long-term study does not really hold true.

Author response: We acknowledge that our study assumes that the overwinter distribution of the population has not changed over time and in the revised manuscript we explicitly state this assumption (lines 231-233). Whereas it is possible that the winter distribution has shifted over time, shifts are likely to have occurred (i) along a north-south axis (as opposed to east-west across the Appalachian Mountains), and (ii) over a shorter distance than that which we sampled winter weather and density. Combined with the fact that the species-wide winter distribution extends from California to Florida and south into Mexico and northern Central America (light purple in map below) geolocator data has substantially refined our understanding of the population-specific winter distribution (royal blue contours in map below) relative to the rest of the species.

Map color legend: purple = winter, light orange = breeding, yellow = migration only, blue-grey = year-round

4. Although the effects of winter mean temperature are clearly visible, it is biologically not clear whether it is mostly

the mean temperature over such a long period that determines survival, or whether it is e.g. a period of harsh conditions. In principal a very mild winter still can have a low survival if there is heavy snow cover for a single week.

Author response: We agree with the reviewer that short periods of harsh conditions could have strong effects on survival during winter and we did, in fact, consider this in our preliminary analyses. In these analyses, we averaged daily temperature and precipitation at monthly intervals as well as over the entire winter period. To summarize the results of these analyses, for all four age-sex groups, average daily temperature over the entire breeding had a greater (or equal) effect on survival compared to temperature in any one month. Results for precipitation were more variable across age-sex groups, but since effects were generally weak (the only monthly effect whose 90% credible interval did not overlap 0 was for adult female survival), we opted to average precipitation across the entire winter period for consistency and simplicity. In addition to average daily temperature, we also considered the proportion of days with temperatures below 0 at monthly intervals and across the entire winter period and, again, for all four age-sex groups, average daily temperatures had greater effects on survival than did the proportion of days with temperatures below 0. Weather measures could be considered at even smaller intervals (e.g., weekly or bi-weekly), but the expected change (if any) would be for an increase in what are already clear limiting effects of winter weather and thus would be unlikely to change the main results of the study. In the revised manuscript, we acknowledge having conducted these preliminary analyses (lines 256-260). We have not included them in the Supplementary Information simply because there were several scenarios analyzed and they are quite lengthy.

5. In general, I am missing a lot of ecological details on the population and the stats, that make the MS difficult to judge. I realise that for a paper in such a journal space constraints are severe, but also in the supplementary material there is rather little info. It is e.g. not clear what (re)capture rates are, whether the models of mark-recapture do substantiate sex*year effects, what the population trend is, whether the local (post-)breeding population and winter density are correlated, whether birds make multiple clutches, etc.

Author response: As suggested, we have added a substantial number of methodological, statistical, and ecological details to the manuscript. These include:

1. **(Re)capture rates.** In the revised manuscript, we report recapture rates for each age and sex group (lines 360-366). Mean recapture/resighting probabilities were high for all four age-sex groups (adult males = 0.96 [0.92, 0.99]; adult females = 0.96 [0.93, 0.98]; juvenile males = 0.93 [0.86, 0.98]; juvenile females = 0.93 [0.87, 0.97]).
2. **Population trend.** In the revised manuscript, we report the temporal trend in breeding population size (lines 67-70).
3. **Relationship between (post-) breeding population and winter density.** As suggested, we investigated the relationship between post-breeding population and winter density and found no evidence for a correlation between the two (lines 95-100).
4. **Double brooding.** In the revised manuscript, we specify proportions of females in each year that successfully fledge at least one clutch and the proportion that successfully fledge two clutches (lines 213-216).

Regarding whether our models of survival substantiate sex*year effects: we recognize the value of model selection for many situations, but we did not conduct such a procedure for the mark-recapture models for a couple of reasons. First, a main goal of our study was to assess the relative contributions of vital rates to variation in population growth rate. Thus, estimating survival by sex and age was necessary to achieve this objective, even if estimates were similar across these categories. Second, given our intensive monitoring protocol, we were afforded the opportunity to specify a population projection model that considered three stages and both sexes. Thus, annual estimates of sex-specific survival were necessary to project stage- and sex-specific abundances.

6. I have some reservation in determining density dependence in populations that show a clear trend over time. Density dependence requires a certain carrying capacity (determined by environmental conditions such as food) and a regulating process that makes the population to grow when being at relatively low density and decline when being relatively high. In case of a population trend, carrying capacity is apparently declining, and it becomes not so

apparent how density dependence would operate and how it can be estimated. The authors do acknowledge this by using detrended breeding population size, but this may not solve the whole problem.

Author response: We agree with the reviewer that detecting and estimating density-dependence can be complicated by temporal trends in vital rates and population abundances. However, presence of temporal trends in population size and/or demographic parameters should not preclude assessments of density-dependence. As the reviewer states, density-dependence requires knowledge of a carrying capacity and population trends signal a shift therein (e.g. a negative population trend signals a decline in carrying capacity at some point in the annual cycle). From a biological perspective, by de-trending population size we account for a declining carrying capacity, such that the density-dependent effect of adding one individual to a breeding population is evaluated in the context of the current (or recent) carrying capacity rather than the carrying capacity during a different point in time. As we highlight on lines 384-386 of the revised manuscript, from a statistical perspective, de-trending helps to avoid spurious detections of density-dependence which may happen if two traits (in our case population size and fecundity) show similar temporal trends (Grosbois et al. 2008). After considering these biological and statistical points, we still detect strong density-dependent effects in this population.

Reference:

Grosbois, V. *et al.* Assessing the impact of climate variation on survival in vertebrate populations. *Biol. Rev.* 83, 357–399 (2008).

It would be interesting also if the authors do mention whether this population decline is a more global pattern, and whether anything is known what causes this.

Author response: As suggested, we have added information regarding the population trend across the range of Savannah sparrows (lines 189-192). Breeding Bird Survey data suggest that Savannah sparrow populations are declining in both Canada and the USA. Average annual percent change of the population was estimated at -1.37 (95% ci = -1.86, -0.913) between 1970 and 2012 in Canada and -1.25 (95% ci = -1.66, -0.90) in the USA between 1966 and 2013. Declines have been most uniform in eastern North America and may be related to urbanization, changing agricultural practices (from dairy to cash crops; Jobin et al. 1996), and forest regeneration (Wheelwright et al. 2008).

References:

Wheelwright, N. T. & Rising, J. D. Savannah sparrow (*Passerculus sandwichensis*). *Birds N. Am. Online* <http://bna.birds.cornell.edu/bna/species/045/articles/introduction> (2008).

Jobin, B., DesGranges, J.-L. & Boutin, C. Population trends in selected species of farmland birds in relation to recent developments in agriculture in the St. Lawrence Valley. *Agr. Ecosyst. Environ.* 57, 103–116 (1996).

Link to BBS results for Canada (<http://www.ec.gc.ca/ron-bbs/P004/A001/?lang=e&m=s&r=SAVS&p=->)

Link to BBS results for USA (<http://www.mbr-pwrc.usgs.gov/cgi-bin/atlas13.pl?05420&1&13&csrfmiddlewaretoken=3YKakk7LxT2ki6NSpl4mstudYCqdW02C>)

7. I find it striking that the authors are not citing one of the best studies on the effects of year-round density on the dynamics of populations, which is of the Icelandic black-tailed godwits by Jenny Gill and co-workers (Gill et al. 2001, Gunnarsson et al. 2005). In their study they knew exactly in which estuary individuals were wintering, and how during population growth sequentially more low quality sites were occupied with lower survival through a density dependent process (which also operates at the breeding grounds).

Author response: We thank the reviewer for pointing out our oversight. We have now cited these papers on lines 92 and 162 of the revised manuscript.

Reviewer #4 (Remarks to the Author):

This manuscript estimates the influence of weather variables and density dependence on demographic parameters of a migratory songbird species. The authors conclude that their study population is regulated by winter temperatures

affecting survival probability, but that increased winter survival can lead to higher population density on breeding grounds which in turn reduces survival of males and fecundity. The manuscript is very well written, and the analyses are both innovative and robust and support the conclusions. Because migrants are experiencing global population declines, the formal framework presented here will allow us to objectively quantify the underlying drivers of population declines for other species, which is a much-needed advancement.

For many declining migrant populations the lack of knowledge about the location of their non-breeding range hampers efforts to assess climatic influences on demographic parameters. In this manuscript the authors use tracking data of birds from their study population to delineate an area over which they obtain winter weather data to ensure that influences of winter weather on demographic parameters are measured in the appropriate region. This approach, and the integrated population model used to estimate demographic parameters, are robust and convincing.

Author response: We thank the reviewer for these very positive comments and constructive feedback.

I found only three issues that would benefit from some clarification:

1) The population count data, which are a critical component of the IPM, are not very well explained. The authors mention territory mapping (L. 155), and that the species is mostly monogamous (L. 166-168), but I think it would be important to specify how each component of the population is counted. In many songbirds we often just count the (singing) males, and simply assume that there are as many females. From the model code it is evident that there are count data for breeding males and breeding females, and it would be good to provide a better description of how the count data were obtained, especially given that some males may 'secure more than one female' (L. 168).

Author response: As suggested, we have added details on how population counts were collected in the revised manuscript (lines 304-311). We have also added a new figure (Fig. 2 in the revised manuscript) showing observed population size and estimated population size side-by-side to demonstrate the consistency between the two measures. We have also included recapture rates to further demonstrate the quality of the data used in the IPM (lines 360-366).

2) An integrated population model works best if the different data sources (count data, mark-recapture data) are independent, but the models are fairly robust even if there is substantial overlap in the data sets (Abadi et al. 2010). From the description of the territory mapping effort it appears likely that the population census data are comprised of counts of color-banded birds, hence the count data may contain exactly the same individuals as the capture-resight data used for the estimation of survival. Although the paper by Abadi et al. (2010) demonstrated that IPMs are robust to violations of the independence assumption, these simulations did not examine estimation of parameters for which no data existed (e.g. immigration), and it would be good to state the extent of the overlap of the two data sets (did unmarked individuals contribute to the population count data?), and how this partial (or complete) non-independence may affect demographic parameter estimates.

Author response: As suggested, we have added additional information to the manuscript describing the extent of overlap of the mark-recapture and population count datasets (lines 304-311). The reviewer is correct in that our population counts are comprised almost entirely of colour-banded birds. However, as mentioned by the reviewer and summarized in the revised text (lines 311-314), Abadi et al. (2010) have shown that estimation of parameters for which data exist within IPMs is robust to violations of independence. Importantly, Schaub & Fletcher (2015) have shown recently that estimation of parameters for which no data exist (i.e., immigration) are also robust to non-independence. Therefore, we are confident that our parameter estimates have not been adversely affected or biased by overlap in our population count and mark-recapture datasets.

References:

Schaub, M. & Fletcher, D. Estimating immigration using a Bayesian integrated population model: choice of parametrization and priors. *Environ. Ecol. Stat.* 22, 535–549 (2015).

Abadi, F., Gimenez, O., Arlettaz, R. & Schaub, M. An assessment of integrated population models: bias, accuracy, and violation of the assumption of independence. *Ecology* 91, 7–14 (2010).

3) I am not familiar with the path analysis, and cannot judge how robust this approach is compared to standard model selection approaches, which are arguably difficult for complex Bayesian models such as this IPM. Because the path analysis is a fairly innovative component, it would be good to briefly highlight the main advantages of this approach over e.g. Gibbs variable selection or the product space method [which may be rather awkward to implement in a complex IPM] - see Tenan et al. 2014. *Ecol. Modell.* 283:62-69.

Author response: As suggested, we have added a brief description of some advantages of the path analysis approach to the methods (lines 393, 396-399). We also highlight some of these advantages in the main text (lines 139-145). These include, the ease in estimating direct effects (of vital rates on population growth rate and of covariates on vital rate) and indirect effects (of covariates on population growth rate), and ensuring proper error propagation by fitting the path model to each posterior sample. The fact that the path model is built using common regression methods has the additional benefit of flexibility and accessibility.

We do want to clarify that path analysis is not a form of model selection *per se*. Rather the path analysis was used to estimate indirect effects of winter and breeding conditions on population growth rate via the vital rates. All variable selection occurred within the integrated population framework, prior to implementing the path analysis. Independent variables included in the path model were selected based on the procedure described in *Methods: weather and density variable selection* (lines 368-391). Briefly, we chose those variables for which the 90% credible interval of the estimated effect deviated from zero the most. We investigated other more sophisticated variable/model selection methods (specifically, the indicator variable method of Kuo & Mallick 1998 as implemented by Koons et al. 2015 as well as Gibbs variable selection as described in Tenan et al. 2014), but, as the reviewer suggests, implementing these methods in an IPM framework proved difficult and thus we opted for a simpler approach.

References:

- Koons, D. N., Colchero, F., Hersey, K. & Gimenez, O. Disentangling the effects of climate, density dependence, and harvest on an iconic large herbivore's population dynamics. *Ecological Applications* (2015).
- Tenan, S., O'Hara, R. B., Hendriks, I. & Tavecchia, G. Bayesian model selection: The steepest mountain to climb. *Ecological Modelling* 283, 62–69 (2014).
- Kuo, L. & Mallick, B. Variable selection for regression models. *Sankhyā: The Indian Journal of Statistics, Series B* 65–81 (1998).

Minor comments:

L. 87-90: This statement is a bit tenuous for females, because the credible intervals of the correlation overlap 0 quite substantially.

Author response: We agree with the reviewer and have re-worded this sentence to better acknowledge that, for females, the relationship between immigration rate and survival was weak (lines 86-90).

L. 179: Although irrelevant for the purpose of this manuscript, there is generally huge concern over the effects of tags on birds. Unless GLS were deployed on juveniles this recovery rate (26%) is a bit lower than the average adult survival probability (~40%, Fig. 2). It would be good to add a comment how you might explain that discrepancy.

Author response: Thank you for making this important point. Geolocators were, in fact, deployed on both juveniles and adults, which explains the discrepancy between the overall geolocator recovery rate and the average apparent survival probability of adults. In the revised manuscript, we explicitly state that geolocators were deployed on both juveniles and adults and present age-specific return rates of individuals to which we deployed a geolocator (see lines 224-233). Age-specific return rates (adult = 0.42; juvenile = 0.25) were very similar to age-specific apparent survival probabilities estimated from the IPM, suggesting that geolocators did not affect survival.

L. 227: It might be good to move L. 260 to here to clarify that $B_{s,t} = R_{s,t} + S_{s,t} + I_t$. Based on the text up to here the recruits and immigrants don't really enter the breeding population, which I found initially confusing (although it is

evident from the model code and the equation in L. 260 that immigrants and recruits enter the breeding population).

Author response: As suggested, we have inserted the formula for calculating time- and sex-specific breeding abundances earlier in the methods describing the IPM to make it clearer how recruits and immigrants enter the breeding population (lines 299-301).

L. 259: Why did you not include this calculation as a derived parameter in the model to ensure adequate error propagation? It is obviously just as valid to use the sims.list output to do that - I'm just curious why it was done outside the model. Were there computational benefits?

Author response: As acknowledged by the reviewer, calculating population growth rate (and immigration rate) inside or outside of the IPM produce the same result and are equivalent in terms of error propagation. Our reason for calculating these parameters outside of the model is because both calculations involve division and the potential exists for a denominator equal to zero at any given MCMC sample. Division by zero will result in an undefined value of the parameter and cause JAGS to crash. Calculating population growth rate and immigration rate after model estimation alleviates this problem.

R-code: Thank you for including the code, which is fantastic to properly understand the model!

Author response: Our pleasure! Thank you again for your helpful reviews of our manuscript.

Steffen Oppel
steffen.oppel@rspb.org.uk

Reviewers' Comments:

Reviewer #3 (Remarks to the Author):

I enjoyed reading this revised version, which I think has clearly improved from adding a lot of relevant ecological details. Most of my previous points have been dealt with appropriately, but there are two points that I feel need some more consideration. These still deal with understanding the ecological processes behind the descriptive patterns observed.

I am surprised that temperatures on the female specific wintering grounds were 5 degrees higher (!), which is more than the difference between the coldest and warmest winter when the whole area is considered. If temperature experienced during winter indeed is the causal effect for explaining the survival, one would expect that male survival would be much lower than female survival. It makes one wonder what is going here biologically, and this is by no means discussed.

I find the discussion not yet very convincing, and making more points about methodology than about the ecological processes involved. (1) I do not understand the argument why DD would be weaker on islands if island habitats are of better quality. DD is a process relative to the carrying capacity, and still one would expect that to work on islands as well, but just at a different level. (2) The paragraph about mismatch does not seem to be very appropriate in the current context, as the only point that is made here is that variation in reproductive success does not contribute a lot to population growth rate. Interestingly, there are effects of density in year t on the survival of adult males and offspring of both sexes to year $t+1$. These demographic rates were more important than reproduction in explaining population growth rate, but one wonders how these patterns come about? There are at least two possible explanations here: (1) High density in year t affects condition of both males and offspring during the breeding season, leading to lower subsequent annual survival, (2) High density in year t is normally followed by above average density in $t+1$, and higher numbers of arriving individuals exclude a larger proportion of individuals to settle. In the first explanation, there seems to be more of an effect of reproduction on growth rate than suggested by the authors, being it not directly measured through number of fledglings, but rather through longer lasting effects. In the second explanation, there must be an additional (or in fact: interactive) effect of winter temperature: in years with lower winter temperatures the effect of population density on annual survival must be lower (because low winter temperatures decrease survival, leading to lower competition, and thereby no individuals are excluded from breeding). The authors do not make a clear attempt in considering these ecological processes, and I find the reasoning for not considering interactive effects of density and winter temperature not very compelling. Their arguments may account for the effects of winter density (and it seems from their response that they do not very much believe that these estimates are biologically reasonable, but if this is so, they should not give them), but the interaction between winter temperature and breeding density still may be relevant. It is not yet clear to me how mild winters would affect the possibilities of local survival through the potential increased competition for limited breeding territories.

The processes suggested about do make the point actually much compelling that a whole annual cycle approach is crucial in understanding population dynamics, not just of migrants but of any

other life-history pattern. I would highly appreciate if the authors do include more ecological processes in their reasoning, rather than stressing that this method is important to actually investigating this annual cycle perspective better.

Reviewer #4 (Remarks to the Author):

The authors have addressed all comments by all reviewers in a thorough and satisfactory manner.

Congratulations on a great paper!

Reviewers' comments:

Reviewer #3 (Remarks to the Author):

I enjoyed reading this revised version, which I think has clearly improved from adding a lot of relevant ecological details. Most of my previous points have been dealt with appropriately, but there are two points that I feel need some more consideration. These still deal with understanding the ecological processes behind the descriptive patterns observed.

Author response: We thank the reviewer for kindly agreeing to evaluate our manuscript for a second time and for, once again, providing constructive and helpful comments.

I am surprised that temperatures on the female specific wintering grounds were 5 degrees higher (!), which is more than the difference between the coldest and warmest winter when the whole area is considered. If temperature experienced during winter indeed is the causal effect for explaining the survival, one would expect that male survival would be much lower than female survival. It makes one wonder what is going here biologically, and this is by no means discussed.

Author response: As suggested, we have added several sentences to the discussion explaining why, despite temperatures being warmer at the female vs. male-specific wintering grounds, annual female survival was like that of males (lines 171-177). One possibility is that other factors (e.g., landscape cover or habitat) are influencing survival differentially across the two sexes. Another point to consider is that similar annual survival between sexes could be the result of lower survival of females during other periods of the annual cycle (e.g., post-breeding, migration, pre-breeding).

I find the discussion not yet very convincing, and making more points about methodology than about the ecological processes involved.

(1) I do not understand the argument why DD would be weaker on islands if island habitats are of better quality. DD is a process relative to the carrying capacity, and still one would expect that to work on islands as well, but just at a different level.

Author response: We agree with the reviewer that density-dependence is a process relative to carrying capacity and that density-dependence is expected to be present in mainland and island populations alike. To alleviate the reviewer's concern, we have removed our example argument for why higher habitat quality could lead to weaker density-dependence on islands (lines 180-191). We have also added a sentence to the discussion explaining, from an ecological perspective, why there may be no or weak density-dependence at the wintering grounds (lines 186-191). In doing so, we further address the reviewer's general concern that the discussion should have a more ecological focus. Finally, please note that we added discussion of why

density-dependence may differ on islands compared to on the mainland to address a comment from Reviewer 1 on the previous revision of this manuscript.

(2) The paragraph about mismatch does not seem to be very appropriate in the current context, as the only point that is made here is that variation in reproductive success does not contribute a lot to population growth rate.

Author response: The point of this paragraph is to emphasize the importance of evaluating the relative contributions of vital rates to population growth rate from both a scientific and conservation perspective. Phenological mismatch is one important way in which environmental change can influence demography, which is why we chose to use it as an example to illustrate our point. We have revised the paragraph to make our point clearer and at the same time lessen the focus on phenological mismatch specifically (lines 152-166).

Interestingly, there are effects of density in year t on the survival of adult males and offspring of both sexes to year $t+1$. These demographic rates were more important than reproduction in explaining population growth rate, but one wonders how these patterns come about? There are at least two possible explanations here: (1) High density in year t affects condition of both males and offspring during the breeding season, leading to lower subsequent annual survival, (2) High density in year t is normally followed by above average density in $t+1$, and higher numbers of arriving individuals exclude a larger proportion of individuals to settle.

In the first explanation, there seems to be more of an effect of reproduction on growth rate than suggested by the authors, being it not directly measured through number of fledglings, but rather through longer lasting effects. In the second explanation, there must be an additional (or in fact: interactive) effect of winter temperature: in years with lower winter temperatures the effect of population density on annual survival must be lower (because low winter temperatures decrease survival, leading to lower competition, and thereby no individuals are excluded from breeding). The authors do not make a clear attempt in considering these ecological processes, and I find the reasoning for not considering interactive effects of density and winter temperature not very compelling.

Author response: We agree with the first explanation proposed by the reviewer that breeding at high densities could affect individual condition and carry-over to influence subsequent survival. We have expanded our explanation in the revised manuscript to make this point clearer (lines 110-115).

We have also added discussion of the second explanation proposed by the reviewer to the revised manuscript (lines 115-122). To evaluate this explanation, we examined the correlation between breeding density in year t and number of surviving adults in year $t+1$. The two were positively correlated (0.56 [0.37, 0.73]), making it possible that the negative effect of breeding density on annual apparent juvenile survival could be through emigration rather than mortality if high numbers of returning adults exclude first-time breeders from settling in the population. However, testing

this prediction will require future work on natal dispersal and age-related settlement patterns during the pre-breeding season. Numbers of local recruits in year t and surviving adults in year t were also positively correlated (0.38 [0.08, 0.64]), suggesting that high density of experienced breeders does not necessarily lead to increased exclusion of first time breeders. Finally, for adult males, it is unlikely that emigration is the source of the negative effect of density on annual apparent survival because breeding dispersal distances of adults are very short (median distance = ~ 30 m) and adults are rarely, if ever, encountered breeding elsewhere on Kent Island or on nearby islands if they have already bred in the study area in a previous year.

Their arguments may account for the effects of winter density (and it seems from their response that they do not very much believe that these estimates are biologically reasonable, but if this is so, they should not give them), but the interaction between winter temperature and breeding density still may be relevant. It is not yet clear to me how mild winters would affect the possibilities of local survival through the potential increased competition for limited breeding territories.

Author response: As suggested by the reviewer, we tested the interaction between winter temperature and breeding density on survival. As revealed by the far right estimate in each panel in the figure below, the interaction had a weak or no effect on annual survival.

The processes suggested about do make the point actually much compelling that a whole annual cycle approach is crucial in understanding population dynamics, not just of migrants but of any other life-history pattern. I would highly appreciate if the authors do include more ecological processes in their reasoning, rather than stressing that this method is important to actually investigating this annual cycle perspective better.

Author response: As suggested, we have included more discussion of our results from an ecological perspective throughout the revised manuscript. Please see responses to specific comments above for line references.

Reviewer #4 (Remarks to the Author):

The authors have addressed all comments by all reviewers in a thorough and satisfactory manner.

Congratulations on a great paper!

All the best,

Steffen Oppel

Author response: We thank the reviewer for kindly agreeing to evaluate our manuscript for a second time and for the positive evaluation of our study.

Reviewers' Comments:

Reviewer #3 (Remarks to the Author):

The authors did a great job in including more ecological reasoning that may underlie the observed patterns. I am fully satisfied with the current MS, which I think is an important contribution to the population ecology of migratory animals.